# Precision Response to the Rise of the SARS-CoV-2 B.1.1.7 Variant of Concern by Combining Novel PCR Assays and Genome Sequencing for Rapid Variant Detection and Surveillance

Nathan Zelyas,[a,b] Kanti Pabbaraju,[c] Matthew A. Croxen,[a,b,g] Tarah Lynch,[c,d] Emily Buss,[a] Stephanie A. Murphy,[e,a] Sandy Shokoples,[a] Anita Wong,[c] Jamil N. Kanji,[b,c,f] Graham Tipples[a,g,h]

[a]Alberta Precision Laboratories, Public Health Laboratory, Edmonton, Alberta, Canada
[b]Department of Laboratory Medicine and Pathology, University of Alberta, Edmonton, Alberta, Canada
[c]Alberta Precision Laboratories, Public Health Laboratory, Calgary, Alberta, Canada
[d]Department of Pathology and Laboratory Medicine, University of Calgary, Calgary, Alberta, Canada
[e]National Microbiology Laboratory, Public Health Agency of Canada, Edmonton, Alberta, Canada
[f]Division of Infectious Diseases, Department of Medicine, University of Calgary, Calgary, Alberta, Canada
[g]Li Ka Shing Institute of Virology, University of Alberta, Edmonton, Alberta, Canada
[h]Department of Medical Microbiology and Immunology, University of Alberta, Edmonton, Alberta, Canada

**ABSTRACT** SARS-CoV-2 variants of concern (VOCs) have emerged as a global threat to the COVID-19 pandemic response. We implemented a combined approach to quickly detect known VOCs while continuously monitoring for evolving mutations of the virus. To rapidly detect VOCs, two real-time reverse transcriptase PCR assays were designed and implemented, targeting the spike gene H69/V70 deletion and the N501Y mutation. The H69/V70 deletion and N501Y mutation assays demonstrated accuracies of 98.3% (95% CI 93.8 to 99.8) and 100% (95% CI 96.8 to 100), limits of detection of 1,089 and 294 copies/ml, and percent coefficients of variation of 0.08 to 1.16% and 0 to 2.72% for the two gene targets, respectively. No cross-reactivity with common respiratory pathogens was observed with either assay. Implementation of these tests allowed the swift escalation in testing for VOCs from 2.2% to ~100% of all SARS-CoV-2-positive samples over 12 January to 9 February 2021, and resulted in the detection of a rapid rise of B.1.1.7 cases within the province of Alberta, Canada. A prospective comparison of the VOC assays to genome sequencing for the detection of B.1.1.7, combined detection of P.1 and B.1.351, and wild-type (i.e., non-VOC) lineages showed sensitivities of 98.2 to 100%, specificities of 98.9 to 100%, positive predictive values of 76.9% to 100%, and negative predictive values of 96 to 100%. Variant screening results inform sampling strategies for regular surveillance by genome sequencing, thus allowing rapid identification of known VOCs while continuously monitoring the evolution of SARS-CoV-2 in the province.

**IMPORTANCE** Different strains, or variants, of severe acute respiratory syndrome coronavirus 2 (SARS-CoV-2, the virus that causes COVID-19) have emerged that have higher levels of transmission, less susceptibility to our immune response, and possibly cause more severe disease than previous strains of the virus. Rapid detection of these variants of concern is important to help contain them and prevent them from spreading widely within the population. This study describes two newly developed tests that are able to identify and differentiate the variants of concern from regular strains of SARS-CoV-2. These tests are faster and simpler than the main, gold-standard method of identifying variants of concern (genome sequencing). These tests also demonstrated a high correlation with genome sequencing and allowed for the rapid and accurate detection of the rise of B.1.1.7 (one of the variants of concern) in the province of Alberta, Canada.

Address correspondence to Nathan Zelyas, nathan.zelyas@aplabs.ca.

New nucleic acid tests for the detection of SARS-CoV-2 variants of concern - these assays detected the rapid rise of B.1.1.7 in Alberta, Canada

Microbiology
Spectrum

**KEYWORDS** B.1.1.7, COVID-19, genome sequencing, PCR assays, SARS-CoV-2, variants of concern

As viruses infect hosts and replicate, they accumulate mutations and evolve over time (1, 2). Accordingly, since the COVID-19 pandemic began, numerous SARS-CoV-2 lineages have been described with various documented mutations (3). While it is anticipated that the majority of these mutations have no biological implications, the emergence of lineages with different phenotypic characteristics has been observed.

To date, there are four variants of concern (VOCs) that have been declared as such by the World Health Organization (WHO): B.1.1.7 (Alpha) (with and without the spike [S] gene E484K mutation), B.1.351 (Beta), P.1 (Gamma), and more recently B.1.617.2 (Delta) (4, 5). Each VOC has a distinct array of mutations in multiple genes, although the S protein is typically the most affected. For example, B.1.1.7 is characterized by a number of mutations in the S gene, including the H69/V70 and Y144 deletions, N501Y, and P681H, among a number of other changes (5). Some of the mutations found in VOCs are postulated to confer a biological advantage for the virus, such as increased transmission or immune escape. The H69/V70 deletion has been demonstrated to increase virus infectivity 2-fold *in vitro* (6). The N501Y mutation (found in all VOCs except B.1.617.2) is found in the receptor-binding domain of the S protein and may confer elevated affinity for its receptor (7). The E484K mutation (found in B.1.1.7 + E484K, B.1.351, and P.1) confers resistance to some neutralizing and commercial monoclonal antibody preparations directed against the S protein (8). Sera from patients immunized with the Moderna or Pfizer-BioNTech mRNA vaccines exhibited a 1- to 3-fold decrease in neutralization activity against B.1.351, which carries both N501Y and E484K, which may indicate a degree of immune escape by this VOC (9).

Recent statistical modeling data indicate increased transmissibility of B.1.1.7, P.1, and B.1.617.2, with B.1.617.2 having an even higher secondary attack rate in household contacts than B.1.1.7 (10–12). Elevated viral loads in specimens positive for B.1.351 suggest the potential for increased transmission (13). Further, data from the UK have led to an estimated 61% higher hazard of death associated with B.1.1.7 infections compared to non-B.1.1.7 infections (14). While generally not present in the currently recognized VOCs, a synonymous mutation in the N gene of SARS-CoV-2 leading to detection failure of that target in a popular commercial assay was found in 18% of sequenced samples in a region in California (15). These findings highlight the need for the timely detection of these variants for their management.

Since the VOCs have been identified, multiple jurisdictions have implemented assays and surveillance systems to rapidly detect them and implement public health and infection control precautions for their containment (16–18). Here, we describe the public health laboratory response, including the VOC screening assay design and implementation, as well as the early detection rates of and rise of VOCs in the Canadian province of Alberta.

## RESULTS

**Performance of the ΔH69/V70 and N501Y assays.** Diagnostic characteristics of the VOC real-time reverse transcriptase PCR (rRT-PCR) assays are summarized in Table 1. The analytical sensitivities using quantified *in vitro* RNA were 2 and 10 copies/reaction (corresponding to 211 and 1,089 copies/ml of sample) for the wild-type and ΔH69/V70 templates, respectively, for the ΔH69/V70 assay. For the N051Y assay, the analytical sensitivities were 3 and 3 copies/reaction (corresponding to 257 and 294 copies/ml of sample) for the wild-type and N501Y templates, respectively. The assays did not react nonspecifically with other pathogens included in the specificity panel, demonstrating 100% analytical specificity. The percent coefficient of variation (%CV) representing assay variability was calculated based on two samples tested in triplicate on three independent runs. For the wild-type and mutant probes for the ΔH69/V70 assay, inter-assay variability ranged from 0.39 to 0.85% and the intra-assay variability

**TABLE 1** Performance of the SARS-CoV-2 variant assays

| Characteristic | Strain or *in vitro* RNA | ΔH69/V70 assay | | N501Y assay | |
|---|---|---|---|---|---|
| | | **Wild-type probe** | **Mutant probe** | **Wild-type probe** | **Mutant probe** |
| Analytical sensitivity[a] | H69/V70 deletion | 2 copies/reaction | 10 copies/reaction | NA | NA |
| | N501Y mutation | NA[b] | NA | 3 copies/reaction | 3 copies/reaction |
| % Analytical specificity[c] | NA | 100 | 100 | 100 | 100 |
| % Inter-assay reproducibility | Wild-type | 0.57–0.85 | NA | 0.91–1.74 | 2.70–2.72 |
| | B.1.1.7 | NA | 0.66–0.74 | 0.96–1.28 | 0.75–0.84 |
| | B.1.351 | 0.39–0.67 | NA | 1.16–2.06 | 0.59–0.66 |
| % Intra-assay reproducibility | Wild-type | 0.31–1.16 | NA | 0.20–1.73 | 0.10–1.88 |
| | B.1.1.7 | NA | 0.08–0.76 | 0.49–1.05 | 0.00–1.04 |
| | B.1.351 | 0.14–0.48 | NA | 0.33–1.23 | 0.13–0.61 |
| % Accuracy | Panel[d] | 98.25 (95% CI 93.8–99.8) | | 100 (95% CI 96.8–100) | |

[a]Analytical sensitivity refers to 95% limit of detection based on probit analysis using *in vitro* transcribed RNA. Numbers for copies/reaction are rounded up.
[b]NA, not applicable.
[c]Analytical specificity based on cross-reactivity to 34 commonly found respiratory pathogens.
[d]Accuracy panel consisted of 10 B.1.1.7 positive samples, 99 wild-type positive samples, and 5 SARS-CoV-2 negative samples.

ranged from 0.08 to 1.16%. For the N501Y assay, inter- and intra-assay variabilities were all less than 3% for both mutant and wild-type probes. Accuracies for the VOC assays using a limited panel of 114 samples, including B.1.1.7 positive ($n = 10$), wild-type positive ($n = 99$), and SARS-CoV-2 negative ($n = 5$) samples demonstrated values of 98.25% (95% confidence interval [CI] 93.8 to 99.8%) for the ΔH69/V70 assay and 100% (95% CI 96.8 to 100%) for the N501Y assay. The cycle threshold ($C_T$) values for the two samples that tested negative for the ΔH69/V70 assay ranged from 33.24 to 40.83 by assays targeting different gene targets, indicating a low viral load.

**Screening the Alberta population for variants of concern.** After 3 February 2021, the described VOC assays were implemented on a large scale and all SARS-CoV-2-positive samples detected in the province were sent to the public health laboratory (ProvLab) for VOC screening. This is reflected in the eventual rise in the 7-day rolling average and proportion of samples screened by the ΔH69/V70 and N501Y assays, from 2.2% of samples on 12 January 2021 to ~100% of samples by 9 February 2021, with consistently high proportions of samples tested for VOCs thereafter (Fig. 1).

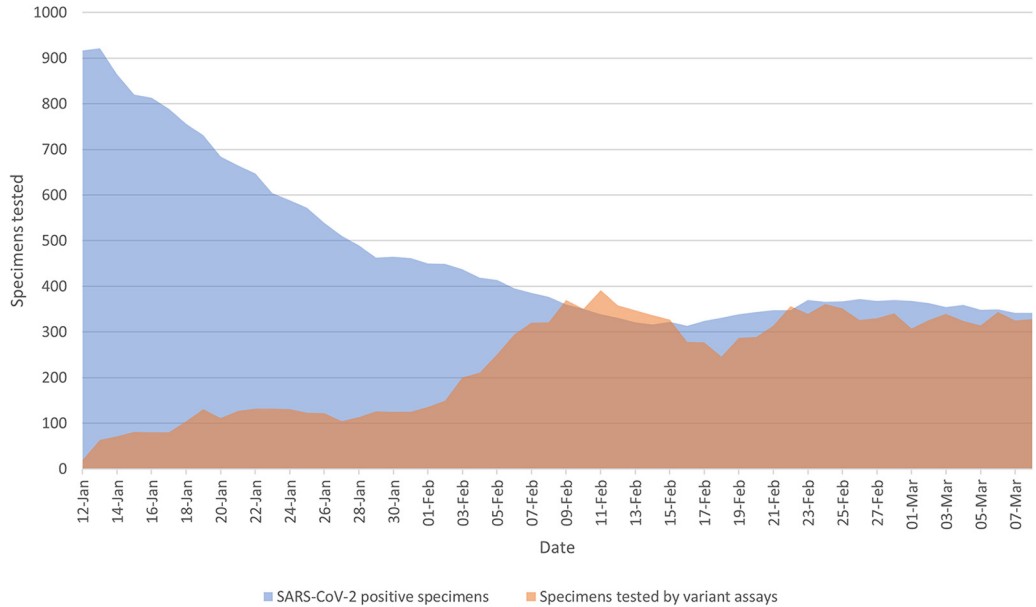

**FIG 1** Rolling 7-day average of SARS-CoV-2-positive specimens tested in early 2021 using the VOC assays. The previous 7-day rolling average of SARS-CoV-2 positive specimens is represented in blue and the same day's previous 7-day rolling average of the number of specimens tested with both the ΔH69/V70 and N501Y assays is represented in orange.

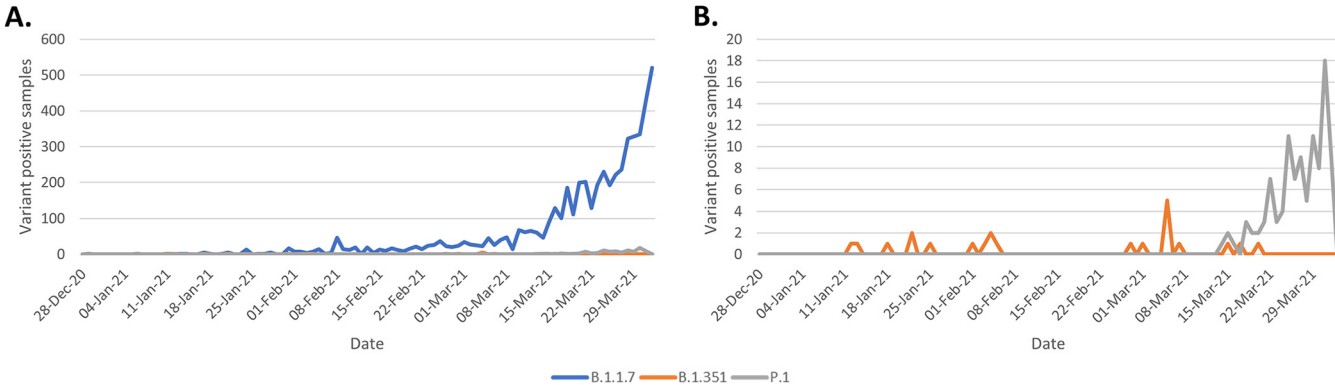

**FIG 2** Daily number of VOC positive samples detected in late 2020 to early 2021. Number of B.1.1.7 (blue line), B.1.351 (orange line), and P.1 (gray line) samples are displayed. (A) Detection of all VOCs. (B) A scaled-down version of the same data to demonstrate the number of B.1.351 and P.1 positive samples.

**Detection of VOCs and comparison with genome sequencing.** Soon after intensive VOC screening was implemented, the rate of B.1.1.7 cases increased markedly (Fig. 2). Early on, screening revealed a B.1.1.7 detection rate of 0.67 to 2.16% (proportion of B.1.1.7 cases detected in a day over a 7-day rolling average of COVID-19-positive cases) in January with a rise to 59.5% on 2 April 2021, corresponding to a total of 5,238 cases identified. In total, 22 B.1.351 and 106 P.1 cases were also detected by 2 April 2021. Results of the VOC assays were compared to genomic results, where the genome completeness was greater than 90% in order to minimize samples with incorrect lineage designation as a result of lower genome coverage, allowing the comparison of results for 3,551 samples. During this time, 585 samples with the H69/V70 deletion alone (without the N501Y mutation) were detected, which were found to be lineages B.1.525 ($n = 574$), B.1 ($n = 5$), B.1.160 ($n = 3$), B.1.258 ($n = 2$), and B.1.280 ($n = 1$). Also during this prospective comparison, a total of 152 samples tested positive for N501Y alone (without the H69/70 deletion); these belonged to the B.1 ($n = 27$), B.1.351 ($n = 17$), B.1.438 ($n = 2$), and P.1 ($n = 106$) lineages.

The performance of the VOC assays when compared to genome sequencing for samples with greater than 90% genome coverage in parallel testing revealed diagnostic characteristics displayed in Table 2. In this analysis, samples positive for both the ΔH69/V70 and N501Y assays were interpreted as positive for B.1.1.7; samples negative for the ΔH69/V70 assay and positive for the N501Y assay were considered presumptively positive for P.1/B.1.351 (essentially non-B.1.1.7 VOCs); and samples negative for both VOC assays were considered wild-type (non-VOCs), since B.1.617.2 was not considered a VOC

**TABLE 2** Prospective comparison of whole genome sequencing with the VOC assays[a]

| VOC assay result | Whole genome sequencing result | | % Sensitivity (95% CI) | % Specificity (95% CI) | % PPV (95% CI) | % NPV (95% CI) |
|---|---|---|---|---|---|---|
| | Positive | Negative | | | | |
| B.1.1.7 | | | | | | |
| Positive | 557 | 0 | 98.2 (96.8–99.2) | 100 (99.9–100) | 100 | 99.7 (99.4–99.9) |
| Negative | 10 | 2,984 | | | | |
| P.1/B.1.351 | | | | | | |
| Positive | 123 | 37 | 100 (97.1–100) | 98.9 (98.5–99.2) | 76.9 (70.7–82.1) | 100 |
| Negative | 0 | 3,391 | | | | |
| Wild-type | | | | | | |
| Positive | 2,832 | 0 | 99 (98.6–99.3) | 100 (99.5–100) | 100 | 96 (94.3–97.2) |
| Negative | 29 | 690 | | | | |

[a]VOC assay result was defined as positive for B.1.1.7 if ΔH69/V70 and N501Y assays were positive, positive for P.1/B.1.351 if only the N501Y assay was positive and the ΔH69/V70 assay was negative, and positive for wild-type if both ΔH69/V70 and N501Y assays were negative. PPV, positive predictive value; NPV, negative predictive value; CI, confidence interval.

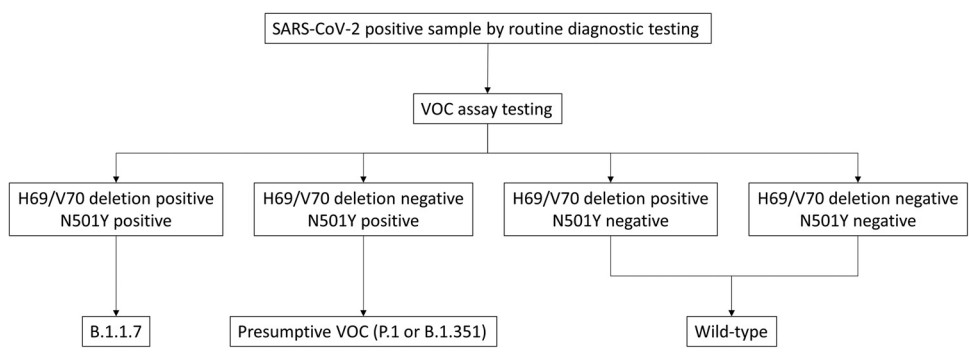

**FIG 3** Interpretation of the VOC assays. Samples testing SARS-CoV-2 positive using routine diagnostic assays were subjected to the VOC assays to classify samples as being positive for B.1.1.7, a presumptive VOC (either P.1 or B.1.351), or wild-type.

during this time period (Fig. 3). Samples that yielded discordant results included eight samples that were misclassified as P.1/B.1.351 rather than B.1.1.7 (all of which were negative on the ΔH69/V70 assay and showed the presence of the C21774T mutation adjacent to the H69/V70 deletion region), two samples that were misclassified as wild-type rather than B.1.1.7 (both of which had the N501Y mutation undetected by the N501Y assay), and 29 samples that were misclassified as P.1/B.1.351 rather than wild-type (all of which showed the presence of the A23063T mutation corresponding to N501Y in the B.1 [$n = 27$] or B.1.438 [$n = 2$] lineages). These discrepancies are summarized in Table 3. The overall numbers of correctly identified VOCs and non-VOC lineages based on the VOC assays compared to genome sequencing are shown in Table 4.

During the study period, the median $C_T$ value for samples tested on the E gene laboratory-developed test with successful VOC assay results was 23.84 ($n = 21,904$). The median $C_T$ value for samples tested with the same diagnostic assay and with successful genome sequencing yielding good quality sequence with at least 90% coverage was 20.86 ($n = 3,394$).

## DISCUSSION

A vital role of the ProvLab is to rapidly respond to emerging pathogens and to develop diagnostics in support of clinical and population health needs. When COVID-19 spread to Alberta in February 2020, the ProvLab developed an in-house SARS-CoV-2 nucleic acid assay and ramped up testing despite the considerable challenges faced by all diagnostic laboratories at the time (19, 20). Almost a year later, a similar challenge was posed by the emergence of variants of concern.

To meet this challenge, assays for the detection of SARS-CoV-2 VOCs were designed and implemented. These rRT-PCR assays, targeting the H69/V70 deletion and N501Y mutation, allowed the rapid identification of B.1.1.7 and other VOCs so that public health contact tracers could rapidly follow up with cases (and contacts of cases) and apply a containment strategy to the variants. The implementation of these VOC screening assays, along with a myriad of other public health mitigation strategies, may have slowed the rise of B.1.1.7 in Alberta, though the enhanced transmission of this variant has likely contributed to its eventual dominance.

**TABLE 3** Summary of discrepancies between the VOC assays and genome sequencing

| Discrepancy | Issue | No. of samples | Lineage by genome sequencing |
|---|---|---|---|
| Misclassified as P.1/B.1.351 | Presence of C21774T mutation adjacent to H68/V70 deletion region | 8 | B.1.1.7 |
| | Presence of N501Y mutation but absence of other VOC-specific mutations | 27 | B.1 |
| | | 2 | B.1.438 |
| Misclassified as wild-type | N501Y mutation undetected by the variant assay | 2 | B.1.1.7 |

**TABLE 4** Lineages determined by genome sequencing and concordance with the VOC assays

| Lineage | VOC assay concordant | VOC assay discordant |
|---|---|---|
| B.1.1.7 | 557 | 10 |
| B.1.351 | 17 | 0 |
| P.1 | 106 | 0 |
| B.1.438 | 1396 | 2 |
| B.1.36 | 405 | 0 |
| A.23.1 | 117 | 0 |
| B.1.1.519 | 128 | 0 |
| B.1.525 | 575 | 0 |
| Others | 211 | 27 |
| **Total** | **3512** | **39** |

The first case of B.1.1.7 in Alberta was detected in late December 2020. As described in our study, it rose to alarmingly high numbers to become the predominant strain by April 2021, highlighting the apparent increased transmissibility of this VOC compared to the concurrently circulating lineages in Alberta. This finding is notable and consistent with other studies of B.1.1.7 in the United Kingdom and United States (21, 22).

While a number of other jurisdictions have relied upon a readily available commercial assay (the TaqPath Thermo Fisher assay) to identify S gene target failures (SGTFs) and rapidly detect potential VOCs, this is an expensive and less-sensitive approach, since B.1.1.7 and B.1.1.7 + E484K are the only VOCs expected to demonstrate SGTFs (16, 18). The combined approach of using both the ΔH69/V70 and N501Y assays described in this report to quickly screen thousands of SARS-CoV-2-positive samples would have detected all VOCs described during the time period of the study and are much less costly, as they are in-house assays that use nonproprietary reagents.

Further, the use of SGTFs to detect VOCs runs the risk of misidentifying a non-VOC strain carrying the H69/V70 deletion, which are known to occur (23). We found 585 samples with the H69/V70 deletion alone without the N501Y mutation, with the majority belonging to the B.1.525 lineage, which is not a VOC but considered a variant of interest in Canada (24). Eight B.1.1.7 positive samples that were misclassified by the VOC assays as potential P.1/B.1.351 positive samples were found to have a mutation adjacent to the H69/V70 deletion region (C21774T), likely being the cause of the negative ΔH69/V70 assay result. However, the wild-type probe for the assay was also negative; thus, a modified interpretation of the assay would have prevented this misclassification by recognizing that identical results for both mutant and wild-type probes is not an expected result for any lineage. While we also found some N501Y positive samples that were not VOCs, these only numbered 29 and the majority of these belonged to the B.1 lineage. The VOC assays described in this study showed a high positive predictive value (PPV) and negative predictive value (NPV) for B.1.1.7 when combined, demonstrating that these markers were sufficient to correctly identify the predominant VOCs currently circulating in Alberta without the need for genome sequencing of these samples. Furthermore, the VOC screening assays have an average turnaround time of 60 h (2.5 days) from sample collection, providing quick results for public health action, compared to approximately 7 days for lineage confirmation using genome sequencing.

Other groups have also implemented rRT-PCR assays to detect mutations associated with VOCs. Wang et al. describe the design and use of assays targeting N501Y, E484K, and L452R, which are all found in different VOCs; this approach allowed them to detect the rise of SARS-CoV-2 strains carrying L452R in the San Francisco Bay Area (25). Very similar to the strategy implemented in our study, another group used commercially available assays to test for the H69/V70 deletion and N501Y in 35,208 SARS-CoV-2-positive samples in France to detect the rise of B.1.1.7 there (26).

This study has several limitations. Prior to 3 February 2021, a large proportion of SARS-CoV-2-positive specimens were not screened for VOCs, and thus it is unclear if

there was already some low-level background circulation of the VOCs in the local population. As well, testing was limited to one province in Canada, decreasing the generalizability and scope of the findings. Because the VOC assays presented only have the N501Y mutation as a marker for B.1.351 and P.1, this led to a low PPV for the VOC assays in detecting these non-B.1.1.7 VOCs (76.7%); thus, samples positive for N501Y alone required further analysis by nucleic acid sequencing, which significantly increases the turnaround time to confirm the lineage result. The assays also have no way of detecting B.1.617.2, as they were designed prior to its emergence, but it has since become an important VOC in Canada. Additionally, the VOC assays lacked an internal control, which is helpful to identify whether inhibitors are present or if extraction was insufficient; however, the fact that the samples were already known to be SARS-CoV-2 positive and included wild-type probes aid in determining if such a technical issue occurs. An important shortcoming of assays targeting only one or two specific mutations in the SARS-CoV-2 genome is that new evolutionary branches from existing VOCs and non-VOC lineages may be undetected. While this is offset in our jurisdiction by performing genome sequencing for the majority of the VOCs detected and a subset of samples testing negative by the VOC assays, the sole use of such screening assays would risk overlooking this far richer source of information.

Overall, the rapid implementation of mutation-specific nucleic acid tests to detect VOCs aided public health in identifying these emerging threats during a time when contact tracing resources were already under significant stress. This work demonstrates that genome sequencing was not necessary to identify characterized VOCs and, in fact, such real-time PCR assays for VOC-specific mutations can allow genome sequencing reagents and resources to be focused on background surveillance to look for new emerging variants. As well, the VOC assays can provide reliable information at lower viral loads than genome sequencing, as indicated by the higher diagnostic assay median $C_T$ value of successful VOC assay results compared to that of successful genome sequencing. Subsequent work will focus on developing multiplex variant assays to identify current and upcoming VOCs more rapidly. Assays that target S gene mutations such as E484K, K417N, K417T, L452R, the Y144 deletion, and the 242 to 244 deletion, and the ORF1ab gene 3675 to 3677 deletion, are in development. As well, methods to increase throughput and reduce turnaround time of genome sequencing are being explored, as it is clear that rRT-PCR assays designed to detect single mutations are effective for identifying currently known VOCs but are limited in detecting new emerging VOCs.

## MATERIALS AND METHODS

**Population and samples.** At the time of writing, SARS-CoV-2 molecular testing in the province of Alberta (population 4.4 million) is being carried out for symptomatic patients and asymptomatic patients who are close contacts of cases or involved in outbreaks. Testing is performed at one of a number of laboratories spread throughout the province (either Alberta Precision Laboratories [APL] or DynaLIFE Medical Labs) or by using a point-of-care nucleic acid or antigen-based test at acute care sites and COVID-19 assessment centers. Upper respiratory tract samples are collected as nasopharyngeal (NP) aspirates or using throat or NP swabs and transported in Universal Transport Medium (COPAN Diagnostics, Remel, Yocon Biology, Phoenix Airmid Biomedical), 0.85% saline (Dalynn Biologicals), or modified liquid Amies (COPAN Diagnostics). Lower respiratory tract samples included sputa, bronchoscopy specimens, or endotracheal tube suctions. Testing data from 25 December 2020 to 2 April 2021 was included in this study. This study was approved by the Human Research Ethics Board at the University of Alberta (reference number Pro00108722).

**Assays for SARS-CoV-2 detection.** Samples testing positive for SARS-CoV-2 at sites across the province were referred to the ProvLab for VOC testing. Assays used to test for the presence of SARS-CoV-2 included the following rRT-PCR assays: E gene laboratory-developed test (19), cobas SARS-CoV-2 (Roche Molecular Systems), Xpert Xpress SARS-CoV-2 (Cepheid), Simplexa COVID-19 Direct (Diasorin Molecular), Allplex 2019-nCoV Assay (Seegene), BD SARS-CoV-2 Reagents for the BD Max System (Becton, Dickinson and Company), Aptima SARS-CoV-2 Assay (Hologic), and an E/N gene laboratory-developed test (unpublished data). Point-of-care testing was also performed using the ID NOW COVID-19 test (Abbott Laboratories) and Panbio rapid antigen test (Abbott Laboratories).

**In-house assays for VOC screening.** Two in-house-designed rRT-PCR assays targeting the H69/V70 deletion and N501Y mutation in the S gene were used for the rapid screening of SARS-CoV-2 positive samples for the detection of potential variants. The primers and probes used are summarized in Table 5.

**TABLE 5** Primers and probes for the detection of the H69/V70 deletion and N501Y mutations in variants of concern for SARS-CoV-2

| Target | Primer/probe name | Primer/probe sequence (5′–3′) |
|---|---|---|
| H69/V70 deletion | Covid_Spk69/70_For | AGTTTTACATTCAACTCAGGACTTGTTC |
| | Covid_Spk69/70_Rev | GACAGGGTTATCAAACCTCTTAGTACC |
| | Covid-Spk6970_WT | CATGCTATACATGTCTCTGG_FAM_MGB |
| | Covid-Spk6970_D | CATGCTATCTCTGGGACC_VIC_MGB |
| N501Y | Covid_SpkN501Y_For | ACACCTTGTAATGGTGTTGAAGG |
| | Covid_SpkN501Y_Rev | AGTTGCTGGTGCATGTAGAAGTTC |
| | Covid-SpkN501Y_WT | CAACCCACTAATGGTGTTGG_FAM_MGB |
| | Covid-SpkN501Y_M | CAACCCACTTATGGTGTTGG_VIC_MGB |

One pair of primers and two probes were designed to detect the presence of wild-type (non-VOC) and mutant sequences in a multiplex reaction. Each SARS-CoV-2-positive specimen was tested by both multiplex VOC assays, referred to as the ΔH69/V70 and N501Y assays. A positive result from the mutant ΔH69/V70 probe with a negative result from the wild-type probe was interpreted as positive for the H69/V70 deletion. A lower cycle threshold ($C_T$) value for the N501Y mutant probe in comparison to the wild-type probe was interpreted as positive for the N501Y mutation. All probes were purchased from Applied Biosystems (ABI, Foster City, California) and primers were purchased from LGC Biosearch Technologies (Petaluma, CA, USA). Both VOC assays were performed using TaqMan Fast Virus One-Step RT-PCR Master Mix (ABI) with 0.8 $\mu$M (each) sense and antisense primers and 0.2 $\mu$M probes combined with 5 $\mu$l of template nucleic acid. The reverse-transcription step was performed at 50°C for 5 min followed by incubation at 95°C for 20 s. Amplification included 45 cycles of denaturation at 95°C for 3 s, followed by annealing, extension, and data acquisition at 60°C for 30 s on the 7500 Fast real-time PCR system (ABI).

**Extraction of viral nucleic acid.** Viral RNA from the different specimen types was extracted on one of two platforms using the manufacturers' instructions: easyMAG (bioMérieux, Marcy-l'Étoile, France) with associated reagents; the MagMAX Express 96 or KingFisher Flex automated extraction and purification systems (Thermo Fisher Scientific) with either the MagMAXTM-96 Viral RNA isolation kit (ABI) or the LuminUltra RNA isolation kit (LuminUltra Technologies Ltd. NB, Canada) in combination with the MagDx AQM magnetic beads from Applied Quantum Mechanics (AB, Canada). Validated specimen types included throat swab, nasal swab, nasopharyngeal swab and aspirate, auger suction, bronchoalveolar lavage, endotracheal secretion, and lung tissue. The sample input and output volumes were 200 $\mu$l and 110 $\mu$l for all the respiratory sample types, respectively, and 60 $\mu$l and 200 $\mu$l for the tissue samples respectively.

**VOC assay analytical sensitivity/specificity, reproducibility, and accuracy.** Regions of the S gene including the targets for the VOC assays were PCR-amplified and cloned into a vector using the TOPO TA Cloning Dual Promoter kit (Life Technologies, CA, USA). The plasmid DNA was linearized using restriction enzymes and the T7 RiboMAX Express (Promega, Madison, WI, USA) or RiboMAX SP6 RNA Production System (Promega, Madison, WI, USA) were used for the transcription of the plasmid DNA to generate *in vitro* RNA. The transcribed RNA was spectrophotometrically quantified for the calculation of copy numbers. The analytical sensitivity for the assay was determined by testing 10-fold serial dilutions of quantified *in vitro* RNA in triplicate on three independent runs. The 95% limits of detection (95% LOD) were calculated by probit analysis. Analytical specificity (cross-reactivity) of the assay was determined by testing a panel of pathogens which included coronaviruses (NL63, OC43, 229E, HKU1, MERS-CoV, and SARS-CoV-1), influenza A (pdm09 H1N1, H3N2, H5, and H7), influenza B, respiratory syncytial virus (A and B), parainfluenza viruses 1–4, rhinovirus 1b, enteroviruses (echovirus 2, coxsackie A16 and B6), adenovirus (types 4, 10, 31, and 40), bocavirus, human metapneumovirus, *Streptococcus pneumoniae*, *Mycoplasma pneumoniae*, *Chlamydia pneumoniae*, *Legionella pneumophila*, *Bordetella pertussis*, *Haemophilus influenzae*, and *Neisseria meningitidis*. The intra- and inter-assay variability were determined using high ($C_T$ value of ~20) and low ($C_T$ value greater than 33) viral load samples for the wild-type, B.1.1.7, and B.1.351 strains, with all samples being tested in triplicate on three independent runs. To assess the accuracy of detection for the ΔH69/V70 and N501Y assays, a panel of B.1.1.7-positive, wild-type-positive (non-VOC), and COVID-19-negative samples were tested by both assays. These samples were identified as belonging to specific lineages based on whole-genome sequence analysis carried out at ProvLab.

**Implementation and interpretation of VOC assays.** Specimens testing positive for SARS-CoV-2 at sites across the province were referred to ProvLab for VOC testing. Those testing positive for the H69/V70 deletion and N501Y were classified as B.1.1.7 and those negative for N501Y were classified as wild-type. Samples testing positive for N501Y alone were initially considered presumptive VOCs (B.1.351 or P.1) and referred for genome sequencing within ProvLab to determine the lineage. These interpretations are described in Fig. 3. As B.1.617.2 (which does not carry the H69/70 deletion nor the N501Y mutation) was not considered a VOC in Canada at the time this study was carried out and its circulation in Canada was undefined (though it was emerging in other parts of the world at the time), it was not included in the VOC assay interpretation.

**Genome sequencing and prospective comparison with the VOC assays.** The full genome of SARS-CoV-2 was amplified by multiplex PCR using the Freed protocol (27) as 1,200-bp amplicons or the Resende protocol (28) and sequenced using Oxford Nanopore or Illumina sequencing technology.

Oxford Nanopore Technologies (ONT) libraries were made using the ARTIC LoCost protocol (29). In brief, the LunaScript Supermix (New England Biolabs [NEB]) was combined with 8 $\mu$l of viral nucleic acid extracted from patient samples for reverse transcription. This was followed by PCR amplification using Q5 High-Fidelity 2X Master Mix (NEB) and 1.1 $\mu$l of 10 $\mu$M primer pools A or B combined with 2.5 $\mu$l cDNA. A total of 35 amplification cycles were performed, including denaturation at 98°C for 15 s and annealing and extension at 65°C for 5 min. The PCR product generated by primer pools A and B were combined and diluted 1:10; Qubit quantification was not performed to improve the turnaround time. End repair for the amplified products was performed using the NEBNext Ultra II End Repair/dA Tailing Module (NEB). Barcoding was done using the NEB Ultra II Ligation Module or Blunt/TA Ligase Master Mix (NEB) followed by cleanup using Spar Q PureMag beads (Quanta Bio) or Ampure XP beads (Beckman Coulter). Adaptors were ligated to the cleaned products using Adapter Mix II from the Native Barcoding Expansion (EXP-NBD104/EXP-NBD114/EXP-NBD196) kit and NEB Next Quick T4 DNA Ligase (NEB). The cleaned libraries were quantified on the Qubit using the dsDNA HS assay kit (Thermo Fisher Scientific) and 15 to 20 ng of the library was loaded on the flo-min 106d flow cells and Ligation sequencing kit (SQK-LSK109) from Oxford Nanopore Technologies (ONT). Alternatively, Illumina libraries were made using the DNA Prep kit (Illumina), and sequenced on an Illumina MiSeq using the 300 cycle MiSeq reagent kit V2 Micro (Illumina).

Consensus genomes from data generated with ONT were completed through the artic 1.1.3 pipeline (https://github.com/artic-network/fieldbioinformatics), while Illumina data was processed with the OICR fork (https://github.com/oicr-gsi/ncov2019-artic-nf) of the ncov2019-illumina-nf pipeline (https://github.com/connor-lab/ncov2019-artic-nf). ncov-qc was used to assess the quality of the sequencing runs (https://github.com/jts/ncov-tools). PANGO lineages (3) were assigned with pangolin (https://github.com/cov-lineages/pangolin).

A subset of COVID-19-positive samples tested by the VOC assays underwent genome sequencing for lineage determination and VOC confirmation. This permitted the clinical sensitivity, clinical specificity, positive PPV, and NPV of the VOC assays to be determined compared to genome sequencing.

**Statistical analysis.** The inter- and intra-assay variability for the VOC assays were determined by calculating mean $C_T$ values, standard deviations, and percent coefficients of variation (%CV) for high and low viral load samples. Accuracies were calculated as (true positives + true negatives)/(true positives + true negatives + false positives + false negatives) × 100%. Previous rolling 7-day averages were determined for specimens testing positive for SARS-CoV-2 and for specimens tested using the ΔH69/V70 and N501Y variant assays. Clinical sensitivity (true positives/[true positives + false negatives] × 100%), clinical specificity (true negatives/[true negatives + false positives] × 100%), PPV (true positives/[true positives + false positives] × 100%), and NPV (true negatives/[true negatives + false negatives] × 100%) of the VOC assays compared to genome sequencing were calculated from prospectively collected samples.

## ACKNOWLEDGMENTS

We are indebted to the ProvLab research and testing staff, as well as the clinical testing laboratories of Alberta Precision Laboratories and DynaLIFE Medical Laboratories for sending samples to ProvLab for variant testing.

We also want to thank the Canadian COVID-19 Genomics Network (CanCOGeN) supported by Genome Alberta and Genome Canada.

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
