## [Reviewer comments · Microbiology Spectrum]

Microbiology Spectrum

Precision response to the rise of the SARS-CoV-2 B.1.1.7 variant of concern by combining novel PCR assays and genome sequencing for rapid variant detection and surveillance

Nathan Zelyas, Kanti Pabbaraju, Matthew Croxen, Tarah Lynch, Emily Buss, Stephanie Murphy, Sandy Shokoples, Anita Wong, Jamil Kanji, and Graham Tipples

Corresponding Author(s): Nathan Zelyas, University of Alberta

Review Timeline:

Submission Date:	May 20, 2021
Editorial Decision:	June 22, 2021
Revision Received:	July 12, 2021
Accepted:	July 15, 2021

Editor: Heba Mostafa

Reviewer(s): Disclosure of reviewer identity is with reference to reviewer comments included in decision letter(s). The following individuals involved in review of your submission have agreed to reveal their identity: Padmapriya P Banada (Reviewer #2)

Transaction Report:

DOI: <https://doi.org/10.1128/Spectrum.00315-21>

June 22, 2021

Dr. Nathan Zelyas
University of Alberta
UNIVERSITY HOSPITAL, 8440 112 ST NW
8440 112 ST NW
Edmonton, Alberta T6G2J2
Canada

Re: Spectrum00315-21 (Precision response to the rise of the SARS-CoV-2 B.1.1.7 variant of concern by combining novel PCR assays and genome sequencing for rapid variant detection and surveillance)

Dear Dr. Nathan Zelyas:

Thank you for submitting your manuscript to Microbiology Spectrum. When submitting the revised version of your paper, please provide (1) point-by-point responses to the issues raised by the reviewers as file type "Response to Reviewers," not in your cover letter, and (2) a PDF file that indicates the changes from the original submission (by highlighting or underlining the changes) as file type "Marked Up Manuscript - For Review Only". Please use this link to submit your revised manuscript - we strongly recommend that you submit your paper within the next 60 days or reach out to me. Detailed information on submitting your revised paper are below.

Link Not Available

Sincerely,

Heba Mostafa

Journals Department
Reviewer comments:

Reviewer #1 (Comments for the Author):

In this manuscript, the authors describe the development of two multiplexed, PCR-based assays to screen for SARS-CoV-2 variant of concern (B.1.1.7, P.1 and B.1.351) by targeting H69/V70 deletion and N501Y mutation. The authors determine the analytical sensitivity and specificity of these assays as well as compare the assay performance to whole genome sequencing (WGS). These assays offer an option for public health laboratories to screen for known variants of concern with a shorter turn-around time. The manuscript is well-written and here are the comments:

It requires high viral load to obtain high quality sequences by WGS. Please indicate the median Ct values of the diagnostic PCR of the samples sent for WGS and compare the sensitivity of the PCR-based assays to the sensitivity of the WGS.

Please include the details of the samples that yield discordant results (10 samples that were negative for B.1.1.7 by typing PCR but positive by WGS, 37 samples that were misclassified as P1/B.1.351 and 29 samples that were misclassified as while-type (table 3)): lineage, Ct values for the diagnostic PCR and the typing PCR, et al.; and comment on the possible cause of the discordance. For the 8 samples (line 285) that were misclassified as P1/B.1.351, were they negative for both Δ H69/70 and wild-type H69/70?

Other groups have used similar approach and developed PCR-cased assays to screen for VOC and VOI. (as examples, DOI: <https://doi.org/10.1128/JCM.00859-21>; <https://doi.org/10.1128/JCM.00913-21>) It is recommended to the authors to cite and comment on these works

Reviewer #2 (Comments for the Author):

The article is well-written and is an important work and much-needed screening test at the time of the worldwide pandemic.

The major concern with the manuscript is the selection of the target genes for VOC screening. 501 is very relevant, however H69/V70 is already present in the US circulating strains since March 2020, and is not considered to be a VOC. the combination of these two is present in only B.1.1.7 (alpha variant) and none other. and targeting this alone would lead to many false VOC positives making it not an ideal gene target for a VOC assay. that could also be a reason for your high discrepancy.

Minor comments.

Page 4, lines 71-73: update the VOCs nomenclature with the latest WHO recommendations (alpha, beta etc.,)

page 5, lines 90-91: considering the emergence of B.1.617 as now the most transmissible variant of concern, please rephrase your sentence to include prior to the emergence of the delta variant or as of a certain date.

page5, line 106: please mention the lab name (you do mention ProvLab later on, but it needs to be mentioned here first).

Page 7, lines 135: it is a good strategy and seems logical. but have you seen analytically if the probes perform at equal efficiency with the same concentration of WT and MT targets? does the

assay always need WT control? the assay is missing an internal control, which is essential for diagnostic assays to reduce reporting false negatives which could result from inhibitors in the samples, improper RNA extraction etc.,

page 9, line 188: B.1.617 is now one of the most important VOCs. I understand at the time of this manuscript preparation it might not have been. but the assays need to be developed with the future emerging VOCs in mind. Therefore, this assay might have limited application and sequencing will always be needed. I recommend, authors update their manuscript and cite this as a limitation of the assay and revise the statement on line 188-190 and correct it as a VOC in other parts of the world, which can potentially enter other countries as well.

The assay would also miss B.1.427 (epsilon/CA variant).

References: please update references

Staff Comments:

Preparing Revision Guidelines

For complete guidelines on revision requirements, please see the Instructions to Authors at [link to page]. **Submissions of a paper that does not conform to Microbiology Spectrum guidelines will delay acceptance of your manuscript.**

Please return the manuscript within 60 days; if you cannot complete the modification within this time period, please contact me. If you do not wish to modify the manuscript and prefer to submit it to another journal, please notify me of your decision immediately so that the manuscript may be formally withdrawn from consideration by Microbiology Spectrum.

If you would like to submit an image for consideration as the Featured Image for an issue, please contact Spectrum staff.

If your manuscript is accepted for publication, you will be contacted separately about payment when the proofs are issued; please follow the instructions in that e-mail. Arrangements for payment must be made before your article is published. For a complete list of **Publication Fees**, including

supplemental material costs, please visit our website.

Response to Reviewers

Please note that line numbers used in the responses refer to the line numbers in the Marked-Up Manuscript – For Review Only version of the manuscript.

1) It requires high viral load to obtain high quality sequences by WGS. Please indicate the median Ct values of the diagnostic PCR of the samples sent for WGS and compare the sensitivity of the PCR-based assays to the sensitivity of the WGS.

Response: This is a very good point. The reviewer is correct that a high viral load is required to obtain good quality sequence by WGS, thus the majority of the samples on which WGS was attempted in the first place had E gene Ct values less than 30 using the ProvLab COVID-19 laboratory-developed test (LDT). The median Ct value noted in samples with successful variant of concern (VOC) assay results during the study period was 23.84 (n=21,904). For the higher viral load samples that were chosen for WGS, the median Ct value for samples that provided a high quality sequence with greater than 90% genome coverage was 20.86 (n=3,394). Please note that only samples for which the Ct values were available by the same assay (ie, the ProvLab COVID-19 LDT targeting the E gene) were included in this analysis. This information was added to the Results (lines 300-303) and Discussion (lines 385-388).

2) Please include the details of the samples that yield discordant results (10 samples that were negative for B.1.1.7 by typing PCR but positive by WGS, 37 samples that were misclassified as P1/B.1.351 and 29 samples that were misclassified as wild-type (table 3)): lineage, Ct values for the diagnostic PCR and the typing PCR, et al.; and comment on the possible cause of the discordance.

Response: the majority of this information is included in lines 290-297. An additional table has been added for ease of interpreting this information (Table 4; line 584).

3) For the 8 samples (line 285) that were misclassified as P1/B.1.351, were they negative for both ΔH69/70 and wild-type H69/70?

Response: This is an excellent point. All eight of these samples were indeed negative for with the wild-type probe of the ΔH69/V70, indicating that there might be an unanticipated mutation in the binding regions of the oligonucleotides used in the assay. This gap in the interpretation of the assay's results is described further in lines 338-344.

4) Other groups have used similar approach and developed PCR-based assays to screen for VOC and VOI. (as examples, DOI: <https://doi.org/10.1128/JCM.00859-21>; <https://doi.org/10.1128/JCM.00913-21>) It is recommended to the authors to cite and comment on these works.

Response: Thank-you for bringing these to our attention. These, along with an additional study describing the use of VOC-targeting assays, have been added to the Introduction (lines 93-95), Discussion (lines 353-359) and References (lines 481-484 and 555-563) sections.

5) The major concern with the manuscript is the selection of the target genes for VOC screening. N501Y is very relevant, however H69/V70 is already present in the US circulating strains since March 2020, and is not considered to be a VOC. The combination of these two is present in only B.1.1.7 (Alpha variant) and none other and targeting this alone would lead to many false VOC positives

making it not an ideal gene target for a VOC assay. That could also be a reason for your high discrepancy.

Response: Thank-you for this comment. Because B.1.1.7 was the most pressing VOC in our jurisdiction at the time, we employed a strategy that would specifically target detecting B.1.1.7 quickly; hence, the H69/V70 deletion and N501Y mutation were selected as targets for the VOC assays. When the H69/V70 deletion was detected alone (ie, in the absence of N501Y), this was considered wild-type. This is described in the Materials and Methods (lines 189-192) but an additional figure has been added (Figure 1) to better demonstrate how the VOC assays were interpreted. The benefit of also including N501Y is that this acts as a marker for potential VOCs that can not be differentiated by the assays (namely, P.1 and B.1.351). However, not having additional markers to identify and differentiate P.1 and B.1.351 are limitations described in the Discussion (lines 365-368).

6) Page 4, lines 71-73: update the VOCs nomenclature with the latest WHO recommendations (alpha, beta etc.)

Response: The WHO designations for each lineage have been added (lines 72-73).

7) Page 5, lines 90-91: considering the emergence of B.1.617 as now the most transmissible variant of concern, please rephrase your sentence to include prior to the emergence of the delta variant or as of a certain date.

Response: The increased transmissibility of B.1.617.2 has been added to the manuscript (lines 88-90).

8) Page 5, line 106: please mention the lab name (you do mention ProvLab later on, but it needs to be mentioned here first).

Response: In-lab SARS-CoV-2 diagnostic testing is carried out by either Alberta Precision Laboratories (APL) (of which ProvLab is a part) or DynaLIFE Medical Labs. Thus, APL and DynaLIFE have been added to lines 110-111.

9) Page 7, lines 135: it is a good strategy and seems logical but have you seen analytically if the probes perform at equal efficiency with the same concentration of WT and MT targets? Does the assay always need WT control? The assay is missing an internal control, which is essential for diagnostic assays to reduce reporting false negatives which could result from inhibitors in the samples, improper RNA extraction etc.

Response: These are excellent points. During our laboratory validation, we found that the CT values for the wild-type N501Y probe were consistently higher than those for the mutant N501Y probe when the mutation was present (and the converse was true in the presence of wild-type N501 sequences) in the accuracy panel and when using in vitro RNA to determine the analytical sensitivity. As well, the high level of inter- and intra-assay reproducibility provided reassurance that variation from this trend seemed unlikely. This led us to the fairly simplistic approach of using a direct subtraction of the wild-type and mutant N501Y probe CT values to classify whether the mutation was present. The comparison of our results with genome sequencing provides further evidence that this approach was sufficient.

In terms of whether a wild-type probe is always needed for the assays, this is assay-dependent due to differing levels of cross-reactivity of the probes. The Δ H69/V70 assay does not necessarily require a wild-

type probe because the mutant probe does not cross-react with the wild-type sequence. However, we found that the N501Y assay did need a wild-type probe due to high levels of cross-reactivity for the wild-type sequence when using the N501Y mutant probe alone. This is precisely why a comparison between the wild-type and mutant probe CT values must be compared to determine whether N501Y is present.

We agree that the lack of an internal control is suboptimal. This has been added as a limitation of the study (lines 370-373). However, the fact that samples were already known to be SARS-CoV-2 positive and the inclusion of wild-type probes aid in the determining if a technical issue occurs.

10) Page 9, line 188: B.1.617 is now one of the most important VOCs. I understand at the time of this manuscript preparation it might not have been but the assays need to be developed with the future emerging VOCs in mind. Therefore, this assay might have limited application and sequencing will always be needed. I recommend the authors update their manuscript and cite this as a limitation of the assay and revise the statement on line 188-190 and correct it as a VOC in other parts of the world, which can potentially enter other countries as well.

Response: We agree that B.1.617.2 has emerged as one of the most important VOCs and have added additional wording to lines 195-196 as suggested. The lack of its detection by the VOC assays has been added as a specific limitation (lines 368-370).

11) The assay would also miss B.1.427 (epsilon/CA variant).

Response: As B.1.427/B.1.429 (Epsilon) is not considered a VOC in Canada nor by the WHO, it was excluded from mention in the study.

12)) References: please update references.

Response: The references have been updated with a number of additional studies.

July 15, 2021

Dr. Nathan Zelyas
University of Alberta
UNIVERSITY HOSPITAL, 8440 112 ST NW
8440 112 ST NW
Edmonton, Alberta T6G2J2
Canada

Re: Spectrum00315-21R1 (Precision response to the rise of the SARS-CoV-2 B.1.1.7 variant of concern by combining novel PCR assays and genome sequencing for rapid variant detection and surveillance)

Dear Dr. Nathan Zelyas:

Your manuscript has been accepted, and I am forwarding it to the ASM Journals Department for publication. You will be notified when your proofs are ready to be viewed.

Sincerely,

Heba Mostafa
Editor, Microbiology Spectrum
